# Binding Networks Identify Targetable Protein Pockets for Mechanism-Based Drug Design

**DOI:** 10.3390/ijms23137313

**Published:** 2022-06-30

**Authors:** Mónika Bálint, Balázs Zoltán Zsidó, David van der Spoel, Csaba Hetényi

**Affiliations:** 1Pharmacoinformatics Unit, Department of Pharmacology and Pharmacotherapy, Medical School, University of Pécs, Szigeti út 12., 7624 Pécs, Hungary; monibalint18@gmail.com (M.B.); zsido.balazs@pte.hu (B.Z.Z.); 2Department of Cell and Molecular Biology, Uppsala University, Box 596, SE-75124 Uppsala, Sweden; david.vanderspoel@icm.uu.se

**Keywords:** ligand, mechanism, pathway, dynamics, channel

## Abstract

The human genome codes only a few thousand druggable proteins, mainly receptors and enzymes. While this pool of available drug targets is limited, there is an untapped potential for discovering new drug-binding mechanisms and modes. For example, enzymes with long binding cavities offer numerous prerequisite binding sites that may be visited by an inhibitor during migration from a bulk solution to the destination site. Drug design can use these prerequisite sites as new structural targets. However, identifying these ephemeral sites is challenging. Here, we introduce a new method called NetBinder for the systematic identification and classification of prerequisite binding sites at atomic resolution. NetBinder is based on atomistic simulations of the full inhibitor binding process and provides a networking framework on which to select the most important binding modes and uncover the entire binding mechanism, including previously undiscovered events. NetBinder was validated by a study of the binding mechanism of blebbistatin (a potent inhibitor) to myosin 2 (a promising target for cancer chemotherapy). Myosin 2 is a good test enzyme because, like other potential targets, it has a long internal binding cavity that provides blebbistatin with numerous potential prerequisite binding sites. The mechanism proposed by NetBinder of myosin 2 structural changes during blebbistatin binding shows excellent agreement with experimentally determined binding sites and structural changes. While NetBinder was tested on myosin 2, it may easily be adopted to other proteins with long internal cavities, such as G-protein-coupled receptors or ion channels, the most popular current drug targets. NetBinder provides a new paradigm for drug design by a network-based elucidation of binding mechanisms at an atomic resolution.

## 1. Introduction

Uncovering the mechanism(s) by which a drug binds to its target is of primary importance in drug design. To date, established experimental methods such as X-ray crystallography [1,2] and cryo-electron microscopy [3,4,5] have been used to capture the atomic resolution structure of a drug bound with its target (called the binding mode). However, these techniques usually do not supply the entire binding mechanism or the intermediate interactions required (the prerequisite binding modes, or PMs), which are often difficult to capture experimentally [6]. Detecting intermediates is especially difficult with targets such as myosin 2 that have long binding cavities. The widely debated ligand recognition, conformational selection, and induced fit mechanisms for ligand binding [7] suggest that the identification of PMs is crucial for a comprehensive understanding of the process.

Recently, molecular dynamics (MD) has emerged as a suitable approach for identifying PMs of specific drugs binding to specific targets [6,8,9,10,11,12]. MD can generate appropriate samples of target–ligand complex structures, allowing conformational flexibility and explicit solvent effects [13,14]. Here, we present an MD-based approach to the study of myosin 2 (a motor protein with a crucial role in eukaryotic motility) and one of its well-known inhibitors. In myosin 2, an ATP bound to the head or motor region is hydrolyzed, which causes conformational changes in the neck region, and this movement is then transferred to actin microfilaments. Myosin 2 is important in muscle contraction, cytokinesis [15,16], the shape formation of cells [17], force generation in cell dynamism [18,19] mitochondrial fission [20], neurite retraction, outgrowth [21], and glioma invasion of the brain [22]. Because of its role in numerous physiological processes, particularly in cellular multiplication and differentiation, myosin 2 has been targeted in several drug design projects [23,24,25], including investigations of cures for breast cancer [26,27] and pancreatic adenocarcinoma [28].

In the past decades, a non-competitive inhibitor of myosin, S-blebbistatin (BS) [29] and derivatives thereof [23,30,31] have been used to increase our understanding of the role myosin plays in fundamental biological processes [22,32,33]. BS was characterized as a selective [34] inhibitor of non-muscle myosin 2, with its inhibitory effect attributed to blocking phosphate release in the force-producing step, which consequently stabilizes the myosin-ADP-P_i_ intermediate [18,24,29] through Switch 1 and the P-loop [4] (Figure 1). A hydrophobic pocket of myosin 2 was also discovered [24] that binds BS (here called “destination”, D in Figure 1). Pocket D is located on the far edge of the long cavity between the bulk interface of the actin-binding cleft and the nucleotide-binding pocket (pocket N). This binding cavity offers long binding pathways up to 20 Å for BS that may associate with several temporary prerequisite binding sites during its migration from BK to pocket D.

In the present study, we introduce a new strategy, NetBinder, to investigate the binding mechanism in the BS-myosin 2 system. NetBinder uses network theory to link the systematically identified PMs and to reconstruct the complete binding mechanism at an atomic resolution. In this manner, we are able to elucidate the complete inhibitory mechanism of BS when binding to the myosin 2 system.

## 2. Results and Discussion

### 2.1. Systematic Mapping of Binding Modes

Mapping the binding pathways of BS up to pocket D requires systematically identifying all possible PMs on the target (internal) surface. The final binding modes of known inhibitors were determined by X-ray crystallography [22,24], and the fast computational docking method (Methods Section) was verified to reproduce these experimental binding conformations correctly (Appendix A). However, it has been shown that a simple series of fast docking calculations cannot deliver a fully systematic mapping [14,35] of all possible binding modes. Therefore, the present NetBinder strategy (Figure 2) applies a systematic search technique called Wrap ‘n’ Shake [35], and PMs were detected by wrapping the entire inner surface of the binding cavity of apo myosin 2 (Figure 1) in numerous copies of BS. The wrapping process resulted in a monolayer of 16 docked conformations of BS covering the entire surface of the binding cavity of myosin 2 (Figure 2, Appendix A). The 16 corresponding complexes formed by the docked BS and myosin 2 molecules were equipped with structural water molecules [14] and further challenged in the shaking steps of sixteen 1 µs long MD calculations in simulation boxes filled with explicit water molecules (Methods, Appendix A).

Shaking accelerated the dissociation of weakly bound ligand conformations [35] by thermal motions of the explicit water bath and target side chains. In the present study, a ligand copy was considered dissociated if the distance (d_D_, Figure 1) between its center of mass and that of the destination BS conformation (in pocket D [24]) became larger than 30 Å. Shaking also allowed an extensive scanning of uncovered segments of the cavity, producing more than 5000 bound conformations for BS, collected in a pool. 

After clustering the pooled contents (Methods Section), 23 PMs were distilled and ranked according to their interaction energies (E_inter_) with the myosin 2 target. Because conformations with low E_inter_ were preferred during clustering (Methods Section), PMs were evenly distributed along the entire cavity (Figure 3A), covering the full range of d_D_ between PM_1_ and PM_23_. The plots of E_inter_ values of the pooled conformations and PMs as a function of d_D_ are shown in Figure 3B and Appendix A. An energy slope was observed in the plot, that is, E_inter_ significantly decreased with a decrease in d_D_. For the PM data points, a linear correlation was calculated for the energy slopes, resulting in a remarkably large squared correlation coefficient (r^2^ of 0.7). This finding is in line with the “energy funnel” concept presented in numerous studies [36,37,38,39], which assumes that a ligand adopts binding positions with decreasing E_inter_ values when approaching the destination (pocket D) in the target molecule. The energy slope obtained is a two-dimensional cross-section of the energy funnel along variable d_D_, representing the position of BS during the binding process. 

There is a “parking lot” (PL, Figure 3C) region of the tunnel gathering the majority of docked conformations centered at a d_D_ of 10 Å (Figure 3B). The PL is practically the next main binding region after the bulk opening (Bulk in Figure 3A) of the tunnel. The PL was shown experimentally (22) to bind halogenated molecules such as pentabromopseudilin (PBP, Figure 1 and Figure 3). The crystallographic binding conformation of PBP overlaps with PM_2_ and PM_4_ in the center of PL (Figure 3C) verifying that the PL is a relevant binding region of BS. Thus, the PL does not differentiate between such ligand conformations but rather serves as a large storage depot before their last steps to pocket D. In PM_1_, BS has a low E_inter_ of −50.1 kcal/mol at a d_D_ of 7 Å (Figure 3B), close to that of pocket D (−46.1 kcal/mol, d_D_ = 0 Å per def.), and therefore, it has a good chance to enter pocket D in one step from PM_1_ (see Section 3 for further discussion). The above findings can be of general importance in mechanism-based drug design. While PL serves as a relatively large storage place, the proximal region at PM_1_ is a narrower transient place during ligand navigation. It may be expected that the ligand can enter pocket D from PM_1_ in a forward step without returning and using bypasses through other PMs backwards. In the search for new druggable sites, the identification of such parking and proximal PMs can be equally important. 

### 2.2. Binding Pathways from Binding Networks

Beyond the knowledge of the single structural snapshot of pocket D provided by Ref. [24], the determination of PMs (Section 1) was necessary to draw the possible binding pathways of BS. The NetBinder strategy approaches the binding process as a networking problem and produces a representative binding pathway (Figure 4A) based on the corresponding network (Figure 4B). Network science has been successfully applied in structural chemistry [40,41,42,43,44,45], and we evaluated whether a network representation of the binding events would simplify the elucidation of the binding mechanism of BS. In the NetBinder network approach, the PMs correspond to nodes and edges, representing segments of ligand pathways by definition. All nodes were considered in light of two attributes, namely d_D_ and the number of edges leading from that node. The first two attributes were adopted from the corresponding PMs of energy slopes (Figure 3B), while the last one was simply counted after the construction of the graph (Figure 4A, Appendix A). Nodes with more than four edges were considered hubs, and interconnected hubs form the backbone of a network. These hubs represent the busiest PMs in terms of ligand binding and constitute a binding pathway of a ligand. A graphic representation of the binding network of BS (Figure 4A) holds all nodes connected by edges and was produced by the conversion of the three-dimensional positions of the PMs (Figure 4B) into a two-dimensional connectivity list (Figure 4A, Methods). 

BS has a complex binding graph with 23 nodes, 47 edges, and 10 hubs (Figure 4A), suggesting various pathways for binding. The central hub PM_3_, has six connections, and the connectivity of the nodes is especially noticeable around the destination (i.e, at PMs with low d_D_ values) and is a direct consequence of the presence of proximal PMs with low E_inter_ values (see also Figure 3B). The network has a massive backbone of ten hubs connected to each other that might serve as an excellent “binding highway” (Figure 4B), and it is clearly distinguishable from sub-nets of peripheral nodes, which can be thought of as anchoring regions or dead ends of ligand migration. BS enters the tunnel via anchoring PMs (PM_22_, PM_18_, PM_15_, PM_12_, and PM_10_) close to the opening to the bulk, and then it enters the PL. There is a variety of possible pathways at the stage of PL. However, some of the PL nodes (PM_6_, PM_4_, and PM_2_) seem essential to arrive at pocket D. Similarly, the next node, PM_3_, is a key hub guiding BS “correctly” towards the destination, as PM_3_ has one direct connection to D and three connections that are only two steps away from D. Two of these indirect links go through the PL nodes PM_2_ and PM_6_, while the third approaches D via PM_1_, which has the lowest E_inter_ of all PMs (see also Figure 3B). The remaining four hubs (PM_5_, PM_9_, PM_11_, and PM_14_) probably belong to a separate dissociative dead end (Figure 4A). The above findings are summarized in a representative binding pathway of BS (Figure 5, Appendix A) based on a massive network backbone with key proximal hubs PM_3_ and PM_1_. This graph-based binding pathway describes the above networking between individual PMs during the full binding process. 

### 2.3. Final Test: Docking to the Destination Pocket 

In the previous sections, NetBinder determined the most important PMs from among the large pools of possible binding conformations. Based on these PMs, a networking approach was used to produce a complete binding mechanism of BS in the myosin 2 tunnel. The binding graph resulted in a representative binding pathway of BS (Figure 4B), leading to PM_1_, which was hypothesized to be a key proximal PM of the lowest E_inter_ (Figure 3B). A validation of PM_1_ remains as a final test of NetBinder. For this, it was investigated if the final crystallographic conformation (D) of BS could be obtained starting from PM_1_ using fully flexible MD simulations without any biasing restraints. 

It is known [24] that for BS to enter into pocket D requires a conformational change to myosin 2, as pocket D is closed by the side chain L262 in the apo form used in the previous sections. During the binding of BS, L262 changes its position (Figure 5), which opens the entrance to the pocket [24]. Because the binding affinity of BS to myosin 2 is rather low (IC_50_, Appendix A) and the above conformational change is also time-consuming, it was expected that a single MD simulation would not dock BS to the destination site. Thus, terminal docking from PM_1_ was attempted in 12 repeated MD simulations with a maximal length of 1 µs each. Two of these simulations resulted in exactly the crystallographic destination position of BS that was determined experimentally [24], with d_D_ values of 0.9 and 1.5 Å, respectively (Appendix A). The atomic level fit of the calculated and crystallographic conformations (Appendix A) demonstrates the precision of MD-based docking and also verified that PM_1_ is indeed a key proximal PM, directly leading to the final crystallographic binding mode D of BS. 

The fully flexible MD simulations (Methods Section) also allowed the detection of (real time) changes in myosin 2 conformation following the docking process at the atomic level. The MD simulation resulting in the best-matching final BS conformation (d_D_ = 0.9 Å) was selected for a detailed analysis and is featured in Figure 6 and Appendix A. The corresponding structural changes during BS docking are listed in Table 1, which also compares the final stage of the simulation with the experimentally determined values.

The docking of BS with myosin 2 starts from an open conformation of the cleft between loops Switch 1 and Switch 2 [46], as represented by the apo structure pre-recovery-stroke conformation [46] used in this study. As BS moves from the PM_1_ starting position, the cleft closes, and a salt bridge forms spontaneously between E459 and R238, which has also been observed in previous computational studies [47,48]. During this process, Switch 2 is pulled by BS into position via a hydrogen bond between its hydroxyl group and the side chain of E459. As BS further proceeds towards the binding site (i.e., as d_D_ decreases), the BS-E459 hydrogen bond disappears and, between 40 and 137 ns, BS flips 90° (Figure 6), which is boosted by an interaction between the carbonyl group of BS and the amide group of Q637, and the carbonyl and hydroxyl groups of BS interacting with E467. The flipped stage is an important prerequisite for BS to fit into its destination pocket. 

Another important driving factor is the hydrophobic interaction of BS with L262, which also moves with BS and makes room in the destination site (Figure 6, Appendix A), which is consistent with its previously reported structural role, and its final position (d_D_ = 1.8 Å) is consistent with its experimentally determined position D [24] (Table 1). The positions of both BS and L262 stabilize after 138 ns for the rest of the simulation. Similar agreements with the experimental position can be observed at residues Y261 and S456 as well. Y261 is involved in a π-stacking between the phenyl group of BS, while S456 is important in myosin 2 isoform selectivity [24]. BS forms hydrogen bonds with G240 and L262, which can also be seen as the MD simulation stabilizes at close to experimental values (Table 1). 

Thus, the above MD docking calculations verified the prediction of NetBinder on PM_1_ and showed that it is a prerequisite binding mode towards crystallographic ligand conformation D.

## 3. Materials and Methods

**Target preparation.** Atomic coordinates of the apo myosin 2 target structure were obtained from the Protein Databank [46] (PDB code 1mmd). Atoms of amino acids missing in the target structure were inserted with Swiss-PdbViewer [49]. Missing terminal and non-terminal amino acids and acetyl and amide capping groups were added with the Schrödinger Maestro program package v. 9.6 [50]. The ADP-P_i_-Mg^2+^ complex was used because it is the intermediate stage of ATP hydrolysis stabilized after BS binds to myosin 2 [18,29]. An ADP-beryllium trifluoride-Mg^2+^ complex was modified to ADP-P_i_-Mg^2+^ by positioning the phosphate ion in the place of the beryllium trifluoride ion (see parametrization of non-amino acid ligands below). The constructed target was then minimized, allowing full flexibility on the heavy atoms (see shaker/energy minimization below). After the minimization steps, target molecule inputs for docking (pdbqt files) were prepared with AutoDock Tools. A united atom representation for hydrogen atoms in non-polar covalent bonds and Gasteiger–Marsili partial charges [51] were applied to the input files.

**Ligand preparation.** The atomic coordinates of the BS and PBP ligands were extracted from PDB structures 1yv3 and 2jhr, respectively. The pKa values of ligand molecules were calculated using the pKa plug-in in Marvin Sketch, v 6.3.0 [52]. Hydrogen atoms were added according to the correct protonation state at pH 7. Energy minimization was performed on hydrogenated structures using the semi-empirical quantum chemistry program package MOPAC [53]. Geometry optimization with MOPAC was carried out with a gradient of 0.001 kcalmol^−1^Å^−1^, and force calculations were carried out with PM3 parameterization. In all cases, the force constant matrices were definitely positive. The minimized ligand molecules were prepared for docking to the targets as described above. 

**Wrapper**. For the BS ligand, the wrapper method [35] was applied to the binding cavity of myosin 2 (instead of the entire surface) by performing 20 consecutive blind docking cycles, which were enough to increase the target–ligand interaction energy close to 0 kcal/mol. In each blind docking cycle, AutoGrid 4.2 was used to generate the grid maps, with boundaries set to cover the whole binding cavity of myosin 2 (for visualization, see Figure 2) using a box of 130 × 100 × 100 grid points centered on the destination BS conformation ([24] PDB 1yv3). The docking cycles were carried out with the AutoDock 4.2 [54] package using the Lamarckian genetic algorithm (LGA). One hundred docked ligand conformations were obtained in each cycle. The docking parameters were used as described in a previous study [55]. Wrapper ended with a trimming step using a cut-off (i.e., a maximum distance between the binding cavity amino acids and the docked conformation) of 3.5 Å. After trimming the BS docking results, only the ligands interacting with the binding cavity amino acids (Appendix A) were retained. These filtering steps reduced the number docked conformations (from a starting value of 87) to 16 conformations, and these were investigated further by molecular dynamics simulations after the prediction of interface water molecules. 

**Prediction of interface water molecules.** Interfacial water molecules play an important role in the dissociating weak binder conformations and improving target–ligand complex structures [35]. Appropriate interface water positions were calculated by MobyWat [56] using the M3 protocol as described previously [42] and also described recently in the HydroDock [14] protocol for docked binding modes. Complexes with the predicted interface water molecules were re-minimized by a two-step minimization algorithm described in the energy minimization section. 

**Parametrization of non-amino acid ligands**. The parameterization of non-amino acid ligands (ADP, P_i_, and BS) was necessary because the AMBER99SB-ILDN force field [57] does not include molecular mechanics parameters for the ligands used in our study. The charge calculation was performed on the R.E.D. Server [58] for the optimized structure (see ligand preparation) with RESP-A1 [58] charge fitting compatible with AMBER99SB-ILDN force fields. The calculations were performed with the Gaussian09 software [59] using the HF/6-31G* split valence basis set [60].

Energy minimization. Target structures were energy-minimized using a two-step protocol including a steepest descent and a conjugated gradient step. The calculations were performed by the GROMACS 5.0.6 [61] software package with the AMBER99SB-ILDN force field [57] and TIP3P explicit water model [62]. The target structure was placed in the center of a cubic box with the distance between the box and the solute atoms set to 10 Å. The simulation box was filled with water molecules and counter-ions to neutralize the total charge of the system. The particle mesh Ewald method was used for long-range electrostatics. The van der Waals and Coulomb cut-offs were set to 11 Å. The convergence threshold of the first step (steepest descent) was set to 10^3^ kJ mol^−1^nm^−2^. In the second step (conjugant gradient) of minimization it was set to 10 kJmol^−1^nm^−2^. The final structures obtained from the energy minimization were extracted for further calculation with wrapper (see wrapper) or subjected to MD simulations (see shaker). 

The molecular dynamics (MD) calculations of various lengths detailed in shaker and final docking were performed with the GROMACS 5.0.6 software package [61] using the AMBER99SB-ILDN force field [57] and the TIP3P explicit water model [62]. The energy-minimized structures were subjected to NPT MD simulations at a temperature of 300 K. For temperature coupling, the velocity rescale algorithm was adopted. Pressure was coupled to the Parrinello–Rahman algorithm with a coupling time constant of 0.5 ps, a compressibility of 4.5 × 10^−5^ bar^−1^, and a reference pressure of 1 bar. A particle mesh Ewald summation was used for long range electrostatics. Van der Waals and Coulomb interactions had a cut-off of 11 Å. 

**Shaker.** Parallel MD runs of a maximum 1 µs each were performed on the 16 BS ligand–target complex structures that were obtained from wrapper and energy-minimized as described above. The MD simulations were performed as described in the molecular dynamics section with the following specific settings. Position and distance restraints were applied in the parallel MD runs as detailed below. Position restraints were applied with a force constant of 10^3^ kJmol^−1^nm^−2^ during the whole MD simulation on the backbone C_α_ atoms of the protein and the heavy atoms of the co-factor (ADP-P_i_-Mg^2+^) and its surrounding amino acids (N127, Y135, K185, T186, and N233). Distance restraints were generated between the atom pairs (Appendix A). A simulation was terminated if ligand dissociation (d_D_ > 30 Å) was observed. The length of the simulations and the d_D_ values calculated for the last frame of the parallel MD runs are detailed in Appendix A. After trimming, 6952 frames were obtained for BS (the “conformation pools”).

**Calculation of intermolecular interaction energy (E_inter_)**. E_inter_ was calculated between myosin 2 target and BS ligand molecules using the Lennard-Jones parameters of the Amber force field [57] in Equation (1).
(1)$$E_{inter}=\overset{N_{T}N_{L}}{\underset{i,j}{\sum }}\left[\frac{A_{ij}}{r_{ij}^{12}}-\frac{B_{ij}}{r_{ij}^{6}}\right]\;A_{ij}=\varepsilon_{ij}R_{ij}^{12}\;B_{ij}=2\varepsilon_{ij}R_{ij}^{6}\;R_{ij}=R_{i}+R_{j}\;\varepsilon_{ij}=\sqrt{\varepsilon_{i\;}\varepsilon_{j}}$$
where ε_ij_ is the potential well depth at equilibrium between the ith (ligand) and jth (target) atoms, R_ij_ is the inter-nuclear distance at equilibrium between the ith (ligand) and jth (target) atoms, N_T_ is the number of target atoms, and N_L_ is the number of ligand atoms.

**Clustering.** The conformation pools were forwarded from the clustering process and E_inter_ for all frames of the conformation pools (the calculation of intermolecular interaction energy). The BS conformations were clustered and ranked by their E_inter_ values and by the closest distance between each heavy atom of the ligand (d_min_). The BS conformation with the lowest E_inter_ value among the cumulated BS copies was selected to represent Cluster 1 (PM_1_). The BS conformation of the second lowest E_inter_ was considered a new Cluster 2 (PM_2_) if d_min_ > d_rnk_, where d_rnk_ was a ranking tolerance (a distance cut-off of separation of clusters (PMs) from each other) set to 1.75 Å. If d_min_ ≤ d_rnk_ for that cluster, then the ligand conformation of the second lowest E_inter_ was placed in Cluster 1. All subsequent clusters were evaluated by this method, and the resulting representative conformations were evenly spread over the myosin 2 cavity without contacting each other. This clustering technique manifests in the shift in the red dots in Figure 3B towards the best binder cluster representative as measured by E_inter_ rather than towards the conformation closer to the destination, pocket D (in other words, the d_D_ of the red dots does not necessarily approach zero). The 1.75 Å distance cut-off was used so that the cluster representatives would not overlap in order to systematically and evenly cover the binding cavity. After clustering, 23 conformations were obtained for BS (Figure 3B, Appendix A). For reference, d_D_ values between the PMs and the DC conformations were also calculated.

**Calculation of distances between the centers of mass**. The distances between the centers of mass (d_D_) of two BS conformations (simulated and reference destination) were calculated for each MD simulation frame and used to eliminate MD frames where a BS copy was dissociated from the myosin 2 surface (Appendix A). The same d_D_ was also used for the calculation of Figure 3B and the network evaluations of the conformation pools. The d_PM_ values were also calculated between the centers of mass of the PMs obtained after clustering, and these values were used to generate the binding graphs (Appendix A).

**Binding network.** The minimal d_PM_ distance between two PMs was taken from the distance matrix (previous point) with a cut-off of d_D_ >12 Å set between PMs (the length of BS is about 12 Å). By this cut-off, BS PM_23_ had no connections to any neighboring PMs. Hence, this PM was not included in the graph generation. Second, graphs were generated with the NetDraw mode of the program MobyWat [42]. This mode of MobyWat was initially used to generate water–water or water–target interaction networks by calculating the distance between heavy atoms. In our case, instead of the distance between oxygen atoms of water molecules, the d_D_ of each PM was calculated to be used as an input. Additionally, in the B-factor column of our input PDB file, the d_D_ measure between the PM and the D center of mass was used. NetDraw’s source code was modified to allow a maximum of ten edges for each node instead of the default maximum of four edges (a legacy from NetDraw’s original use for water networks). Gephi 0.9.2 [63] and CorelDraw were used to visualize and re-draw the graphs according to the structural information of the PM positions and the edges generated by NetDraw.

**The final docking** of BS was performed by twelve simulated annealing, fully flexible MD runs of 1 µs each (Appendix A) using the same MD parameters as described above, except that simulation annealing was implemented with the temperature scheme presented in Appendix A. The target was fully flexible in these runs, no position restraints were applied at all, except that the cofactor and frames were exported every 0.1 ns, resulting in 10^4^ frames after 1 µs of simulation. The d_D_ of BS was calculated for each frame during the 1 µs (Figure 6). For the d_D_ of the BS conformation in the final frame, see Appendix A.

## 4. Conclusions

Experimental structure determination techniques provide invaluable information of the atomic resolution binding modes of drugs to their macromolecular targets. While some techniques for determining binding dynamics have been suggested [64], complete drug mechanisms cannot be produced routinely, and a systematic method for detecting prerequisite binding modes (PMs) has been lacking. Based on the static structures from experiments, theoretical methods have provided the missing binding dynamics, and some PMs of drugs have been found [6,10,12,35] that should be considered “footsteps” on a drug’s pathway towards its destination on the target. However, the connections between these PMs have hitherto not been uncovered. NetBinder solves this problem by creating networks of PMs and extracting binding pathways. NetBinder, combined with fully flexible MD simulations for the final docking stages, can supply complete mechanisms for drugs that target a long tunnel-shaped binding channel, such as that of myosin 2. Similarly shaped binding channels often occur in transporters [65], important receptors such as muscarinic [66], enzymes such as cyclooxygenase-1 and -2 [67], and transmembrane viral ion channels, such as that of the influenza A [68,69] and SARS-CoV-2 viruses [14,70], and it is our hope that the NetBinder strategy presented here will be adapted to aid in the investigation of such targets. The present work also offers a starting kit of new tools for the identification and classification of PMs, binding sites (parking and proximal), and networking elements (hub and backbone) in mechanism-based drug design.

## Figures and Tables

**Figure 1 ijms-23-07313-f001:**
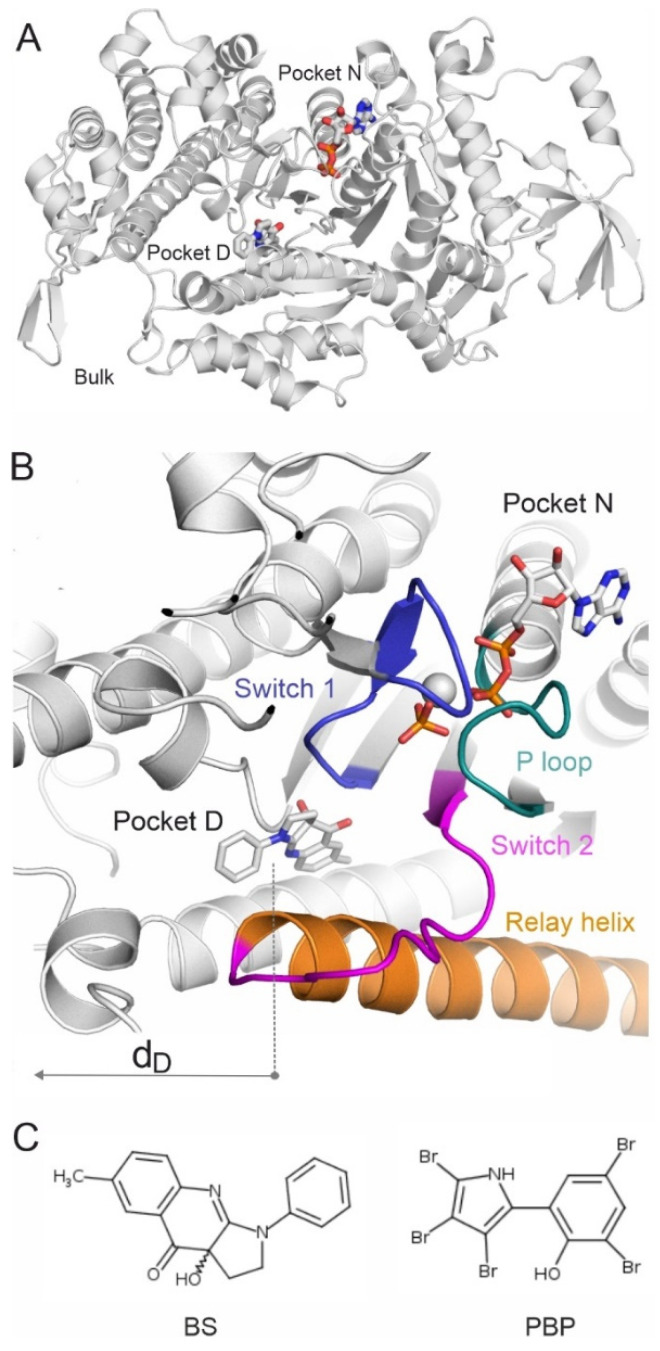
(**A**) Myosin 2 is shown in grey cartoon. The destination pocket (Pocket D) and nucleotide-binding pocket (Pocket N) are highlighted by the experimental binding positions of BS and ADP (grey sticks), respectively. The figure was prepared using the holoenzyme structure (PDB code 1yv3). (**B**) A close-up of the far edge of the binding tunnel inside myosin 2 including pockets D and N with key structural elements in different colors. The switches, the P-loop, and the relay helix are important elements of the enzyme mechanism and also indicate the location of pocket D. The distance of center of mass (d_D_, arrow) of a binding mode of BS is measured from the center of pocket D. (**C**) The Lewis structures of BS and PBP.

**Figure 2 ijms-23-07313-f002:**
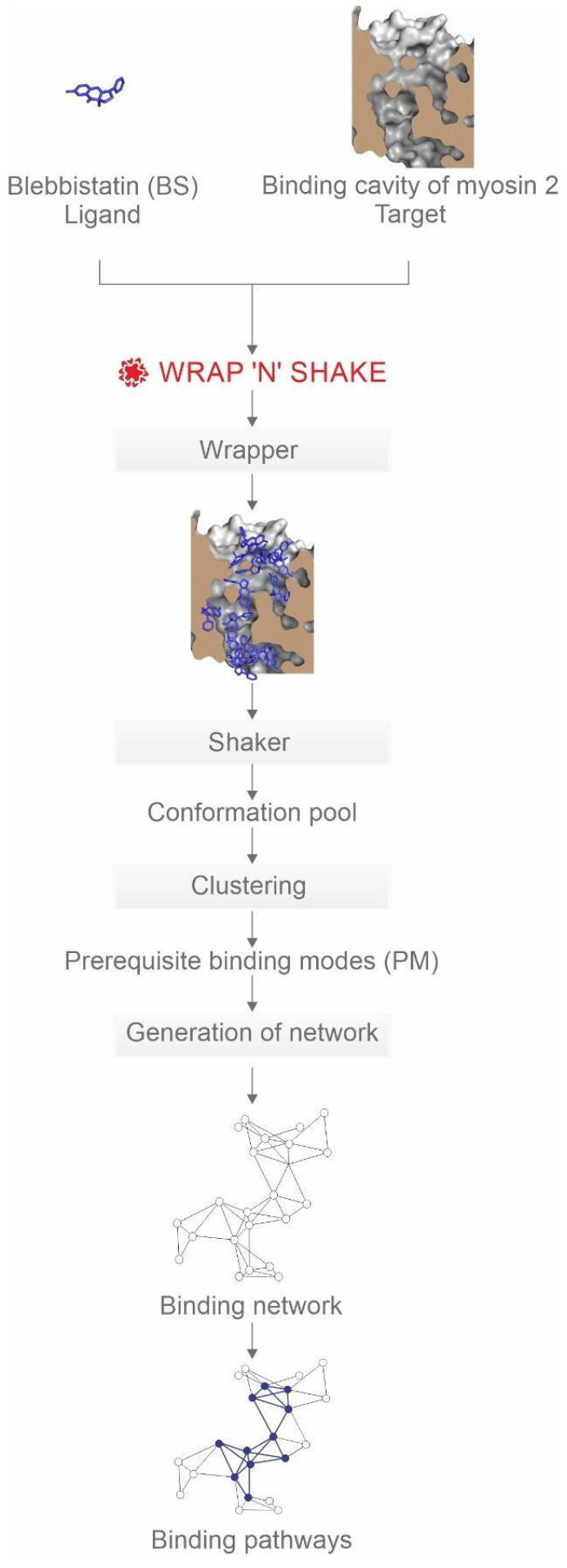
The NetBinder strategy with BS as a ligand and myosin 2 as a target.

**Figure 3 ijms-23-07313-f003:**
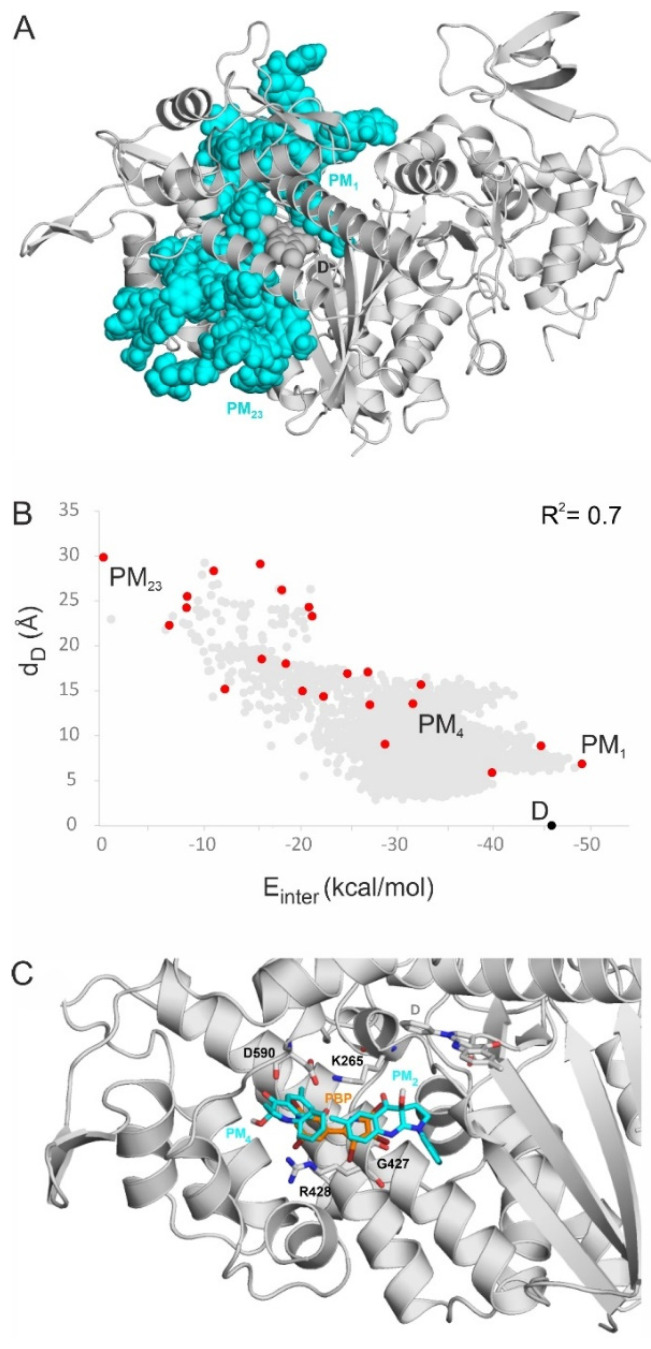
(**A**) The BS PMs (teal spheres) cover the entire binding cavity of myosin 2. The myosin 2 protein is shown as a grey cartoon. PMs with the smallest (PM_1_) and largest (PM_23_) d_D_ values are labelled. BS in pocket D (from PDB 1yv3) is shown as grey spheres. (**B**) Interaction energies (E_inter_) calculated between the PMs of BS and myosin 2 correlate with d_D_ (energy slope in red dots). A similar trend can be observed for the raw BS conformations of the pools (grey dots). The PMs are ordered by the distances of their centers of mass, measured from that of the crystallographic destination (D) binding mode. (**C**) The overlapping binding position of PBP (from PDB 2jhr) and PMs 2 and 4 from the present study. BS bound to pocket D (from PDB 1yv3) is highlighted as grey sticks. PM2, 4, and PBP are shown as teal and orange sticks, respectively. Surrounding amino acids that participate in the binding of PBP, according to the 2jhr structure [25],^,^ are shown as grey sticks and are labelled accordingly. Non-polar (C-connected) H atoms are not shown for the calculated ligand molecules (PM) in the figures for clarity.

**Figure 4 ijms-23-07313-f004:**
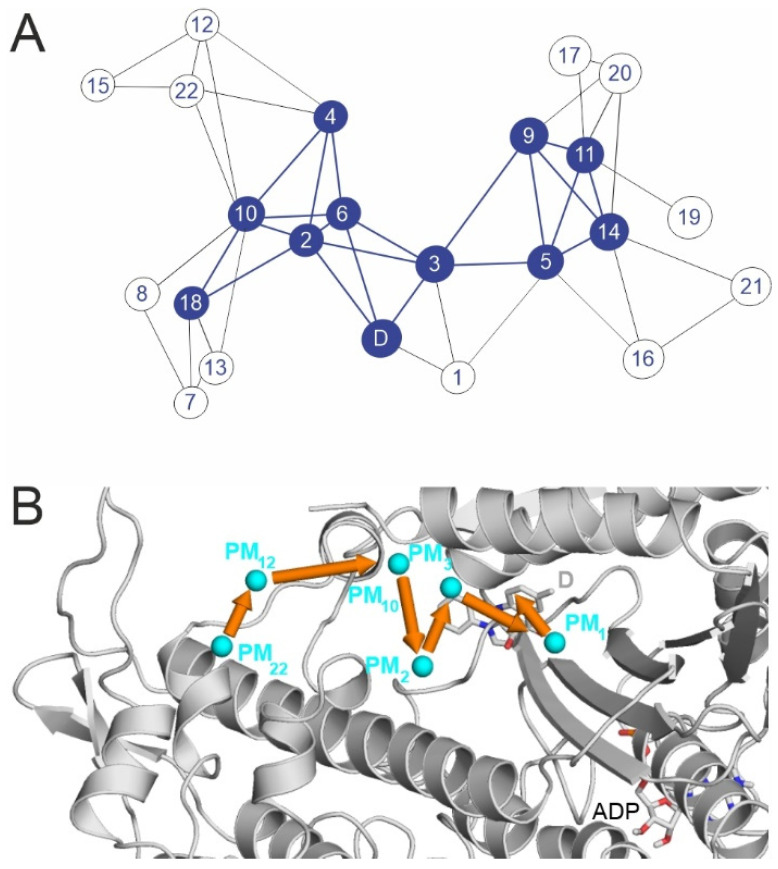
(**A**) Binding network of PMs as nodes for BS. Simple nodes are empty circles; hubs are blue full circle plates. Edges are black lines, and backbone edges are highlighted as blue lines. (**B**) A suggested binding pathway of BS, see also Appendix A. The myosin protein is shown as a grey cartoon. ADP molecule is shown with grey all-atom representation sticks. PMs are shown as blue spheres, and their movement is highlighted with orange arrows. The experimental binding mode (D, superimposed from PDB 1yv3) is shown with grey sticks.

**Figure 5 ijms-23-07313-f005:**
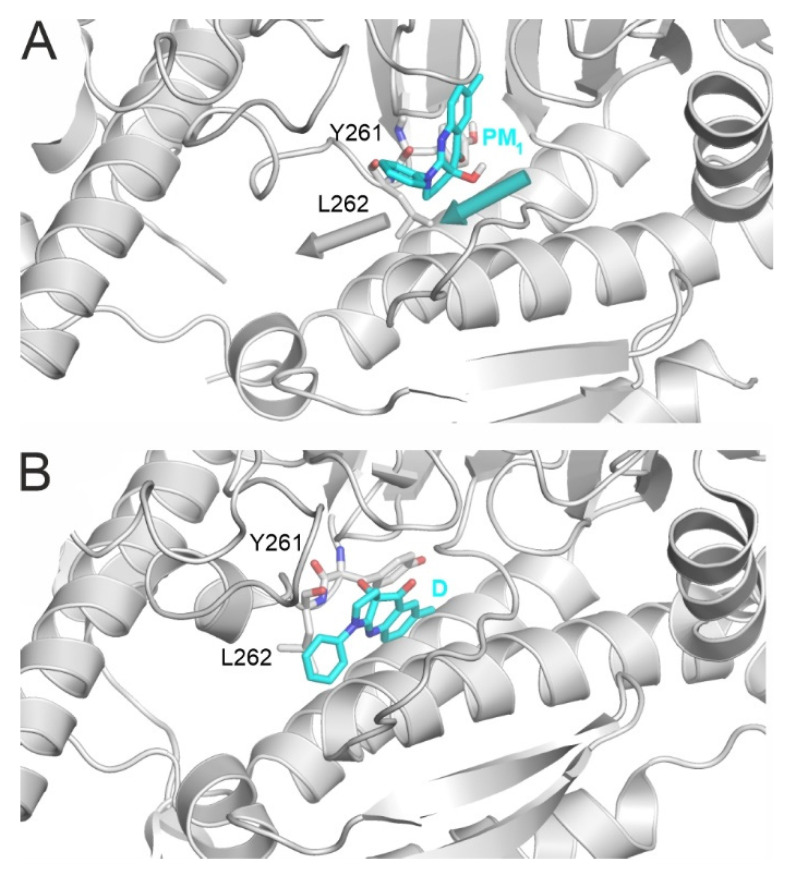
(**A**) In the apo myosin structure (top), L262 blocks the entrance of BS from PM_1_ (d_D_ = 6.9 Å) towards the holo conformation bound to pocket D (**B**). Myosin is shown with grey cartoon. The important amino acids are highlighted with grey all-atom representation sticks and are labelled accordingly. BS is shown with teal all-atom representation sticks. Arrows indicate the movement of L262 and BS.

**Figure 6 ijms-23-07313-f006:**
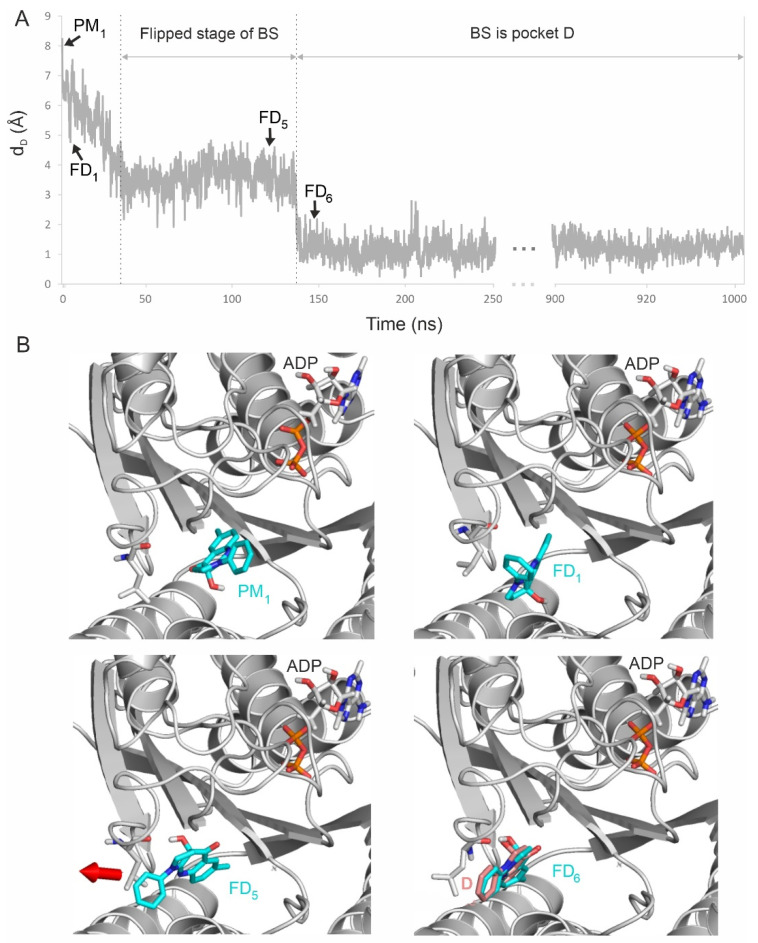
(**A**) Final docking (FD) of BS starting from PM_1_ (t = 0 ns) to the destination pocket, represented by a continuous decrease in d_D_ during the 1 µs simulation. (**B**) Main FD events of the binding process are highlighted (see also Appendix A for all six final docking steps) as well as the corresponding snapshots of BS and the surrounding amino acids of myosin 2. First, BS undergoes flipping, where its phenyl ring (in FD_1_) turns towards L262 to form a hydrophobic interaction in position FD_5_. Then, the inward movement of L262 pulls (red arrow) BS towards the destination binding mode into FD_6_, which agrees well with reference D (pink sticks) from PDB 1yv3.

**Table 1 ijms-23-07313-t001:** Structural changes during final docking of BS into the destination site.

Structural Change		Starting Time (ns)	Distance (Å)		
			0 ns	1000 ns	Experimental
Formation of H-bond (G240…BS)		40	8.3 ^a^	3.2 ^a^	2.8 ^a^
Formation of salt bridge (E459…R238)		71	8.3 ^b^	4.0 ^b^	4.2 ^b^
Movement of BS		137	8.0 ^c^	0.9 ^c^	0 ^c^
Formation of H-bond (L262…BS)		137	6.6 ^d^	2.5 ^d^	2.5 ^d^
Flipping of L262		137	3.4 ^c^	1.8 ^c^	0 ^c^
Flipping of Y261		143	1.9 ^a^	1.4 ^a^	0 ^a^
Flipping of S456		447	4.87 ^a^	1.9 ^a^	0 ^a^

^a^ Distance between NH (G240)—OH (BS); ^b^ Distance between CZ (R238)—CD (E459); ^c^ d_D;_
^d^ Distance between O (L262)—OH (BS).

## Data Availability

Files are provided online at: https://zenodo.org/record/6536287#.YnzSsFRByHt. Accessed on 27 June 2022.

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
