# Peer review of "Binding Networks Identify Targetable Protein Pockets for Mechanism-Based Drug Design"

_ijms, 2022, doi:10.3390/ijms23137313_

Round 1
Reviewer 1 Report
Dear Authors,
This manuscript by Mónika Bálint and co-workers presents a new tool that can be useful in the drug design process. In my opinion, this topic will be of interest to a wide range of readers who work with the theoretical approach to drug design. This manuscript is written in a clear and understandable way and shows new and interesting studies. However, I have some points that must be addressed before the paper will be suitable for publication in the International Journal of Molecular Sciences. So, this manuscript should be accepted after minor revision.
In details: The English language should be carefully checked again. There are minor errors in the manuscript, e.g.(1) on line 19 "to" should be removed. There is "...and to uncover the entire binding mechanism..." and it should be "...and uncover the entire binding mechanism..." (2) on line 63: There is “…Pocket D is located in…” and it should be “…Pocket D is located on…” (3) in line 65: there is “…may associate to several…” and it should be “…may associate with several…” (4) on line 70: there is “…using the holo enzyme structure…” and it should be “…using the holoenzyme structure…” (5) on line 431: there is “…these PMs has hitherto…” and it should be “…these PMs have hitherto…”
Author Response
This manuscript by Mónika Bálint and co-workers presents a new tool that can be useful in the drug design process. In my opinion, this topic will be of interest to a wide range of readers who work with the theoretical approach to drug design. This manuscript is written in a clear and understandable way and shows new and interesting studies. However, I have some points that must be addressed before the paper will be suitable for publication in the International Journal of Molecular Sciences. So, this manuscript should be accepted after minor revision.
In details: The English language should be carefully checked again. There are minor errors in the manuscript, e.g.(1) on line 19 "to" should be removed. There is "...and to uncover the entire binding mechanism..." and it should be "...and uncover the entire binding mechanism..." (2) on line 63: There is “…Pocket D is located in…” and it should be “…Pocket D is located on…” (3) in line 65: there is “…may associate to several…” and it should be “…may associate with several…” (4) on line 70: there is “…using the holo enzyme structure…” and it should be “…using the holoenzyme structure…” (5) on line 431: there is “…these PMs has hitherto…” and it should be “…these PMs have hitherto…”
The Reviewer is acknowledged for his/her careful work and suggestions.
„To” is removed from line 19, the text now reads as: „… and uncover the entire binding mechanism…”.
On line 63 „…Pocket D is located in…” is changed to „…Pocked D is located on…”.
In line 66 „…may associate to several…” is changed to „…may associate with several…”.
On line 71 „…using the holo enzyme structure…” is changed to „…using the holoenzyme structure…”.
On line 433 „… these PMs has hitherto…” is changed to „…these PMs have hitherto…”.
English language at other parts of the manuscript was checked again.
Reviewer 2 Report
Identifying binding sites is extremely important to validate proteins as targets in the discovery of new drugs, since many structures are deposited as apo structures. Thus, the topic of this article is highly relevant and of interest, since much of the information can be applied to the study of several protein classes. The work Binding networks identify targetable protein pockets for mechanism-based drug design describes well a methodology for searching these sites. The work is well-written. However, the manuscript describes the methodology only for myosin 2, which I see as a limitation of the work. Although computational computation is high for these analyses, I believe the work would be more robust if other targets were included for validation. Otherwise, the title of the work could be modified, since it generalizes a methodology that was tested only against myosin 2.
Other minor comments are as follows:
On figure 3. Why are you using the ligand with and without hydrogen atoms?
Page 11 line 267 – I found some PDB ID with best resolution for myosin studied. Why are you don’t use these ID?
Author Response
Identifying binding sites is extremely important to validate proteins as targets in the discovery of new drugs, since many structures are deposited as apo structures. Thus, the topic of this article is highly relevant and of interest, since much of the information can be applied to the study of several protein classes. The work Binding networks identify targetable protein pockets for mechanism-based drug design describes well a methodology for searching these sites. The work is well-written. However, the manuscript describes the methodology only for myosin 2, which I see as a limitation of the work. Although computational computation is high for these analyses, I believe the work would be more robust if other targets were included for validation. Otherwise, the title of the work could be modified, since it generalizes a methodology that was tested only against myosin 2.
The Reviewer is acknowledged for his/her careful work and suggestions. We agree that involving other systems may be of interest. Indeed, the computational cost is relatively high, and the analysis was restricted to the myosin 2 binding cavity. However, due to the length and complexity (branching) of the myosin 2 cavity, it can be considered as a most difficult system, applicable for testing various prerequisite binding situations and their validation. At the same time, experimental information for validation of prerequisite binding pockets is rather limited for other targets. In the case of myosin 2, such information was available for comparison with our calculation results. In agreement with the suggestion of the Reviewer, we modified a sentence in the abstract emphasizing the importance of involving other systems in future studies applying NetBinder. (lines 25-27)
Other minor comments are as follows:
On figure 3. Why are you using the ligand with and without hydrogen atoms?
Non-polar (C-connected) H atoms are not shown for the calculated ligand molecules (PM) in the figures for clarity. This statement was also inserted at lines 131-132 in the caption of Fig. 3. Ligand PBP (orange) is taken from the experimental X-ray crystallographic structure (PDB: 2jhr), in which there are no hydrogen atoms.
Page 11 line 267 – I found some PDB ID with best resolution for myosin studied. Why are you don’t use these ID?
The structure used in the present study (PDB: 1mmd) was also used by other, eminent studies (e.g. ref 47 of the main text) of the field. This 2 Å resolution structure was found to be appropriate starting point for our MD simulations, as well. Moreover, the binding tunnels of similar other structures with slightly lower resolutions are practically the same as that of 1mmd. As the present study focuses mainly on the binding tunnel, benefits from using other structures are not expected by the authors.